# PhotoGraph: Claim-Centric Knowledge Graphs for Personal Photo Management

Omair Shahzad Bhatti[1], Abdulrahman Mohamed Selim[1], László Kopácsi[1], Michael Barz[1] and Daniel Sonntag[1,2]

[1]*Interactive Machine Learning, German Research Center for Artificial Intelligence (DFKI), Germany*

[2]*Applied Artificial Intelligence, University of Oldenburg, Germany*

## Abstract

Personal photo collections are increasingly organised with machine learning, yet many systems still cannot answer contextual questions in an inspectable way. Embedding-based retrieval supports natural language search, but provides weak support for relational constraints, visual evidence, provenance, and persistent correction. We present PHOTOGRAPH, a claim-centric, spatially and temporally aware knowledge graph framework for personal photo understanding and photobook co-creation. PHOTOGRAPH represents model outputs as evidence-grounded claims with lifecycle state, enabling contextual retrieval with justifications and event-based creation with fact-grounded storylines. A video showcasing the tool can be found here.

## Keywords

knowledge graphs, personal photo management, image retrieval, human-AI collaboration, visual storytelling

## 1. Introduction

Personal photo collections have become large, heterogeneous, and semantically complex, often spanning thousands of images across multiple years [1, 2, 3, 4]. Machine learning driven photo management systems aim to make these collections searchable using object detection, face recognition, scene classification, and image search (e.g., Immich[1] and Google Photos[2]). Still, users often cannot get clear answers to *who did what, where, and when.* In many systems, retrieval is based on embedding similarity, sometimes combined with simple tags [5]. This works for broad queries, but it is unreliable for compositional requests that require multiple constraints. For example, *"Marie wearing a black jacket at the beach"* combines a person, an object with an attribute, a relation, and a scene. If these parts are not represented explicitly, the system cannot show which constraints were met, which were missed, or what visual evidence supports the result [6].

Motivated by the limitations of tag- and embedding-based image retrieval, research has explored structured semantic representations, such as scene graphs, relation-centric vision representations, and event models, to support retrieval and reasoning [7, 8, 9]. More broadly, multimedia retrieval work has shown that users ask semantic and contextual questions that sparse tags cannot answer reliably [10]. This gap is especially visible in photobook creation, where meaningful narratives depend on events and interactions; who appears, what happens, where, and in what order matter more than isolated labels [11, 12]. However, many Human-AI co-creation pipelines still provide limited provenance and grounding and offer few mechanisms for revision, which makes outputs hard to inspect, improve, and reuse.

Therefore, retrieval research has moved beyond single-shot text search toward interactive and multimodal querying. Composed image retrieval (CIR), for instance, retrieves images using a reference image together with a textual modification [5]. Recent CIR systems increasingly rely on vision-language

*GenAIK-NORA: The Joint Workshop on Generative AI and Knowledge Graphs and Knowledge Graphs & Agentic Systems Interplay, co-located with IJCAI-ECAI 2026, August 17, 2026, Bremen, Germany*

✉ omair_shahzad.bhatti@dfki.de (O. S. Bhatti); abdulrahman.mohamed@dfki.de (A. M. Selim); laszlo.kopacsi@dfki.de (L. Kopácsi); michael.barz@dfki.de (M. Barz); daniel.sonntag@dfki.de (D. Sonntag)

[1]https://immich.app/ (Accessed February 13, 2026)

[2]https://photos.google.com/ (Accessed February 13, 2026)

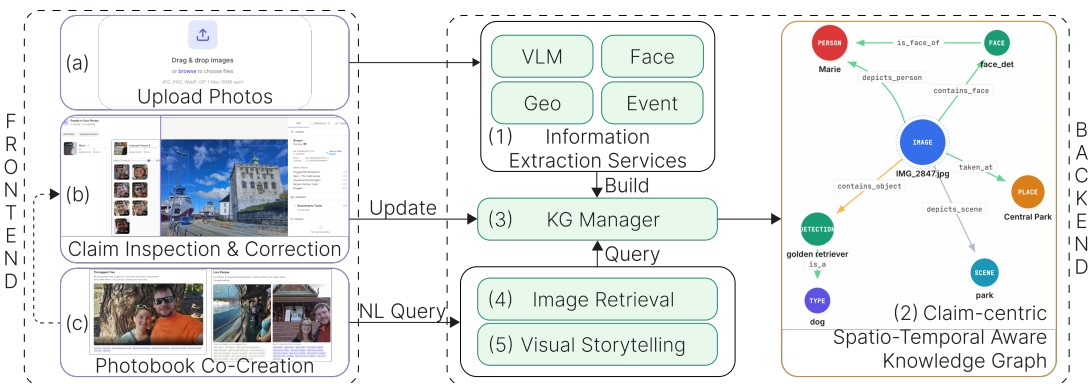

**Figure 1:** PHOTOGRAPH overview: information extraction services (1) produce claims which the KG Manager (3) organises into a claim-centric spatio-temporal knowledge graph (2), enabling contextual retrieval (4) and visual storytelling (5) through natural-language queries.

model (VLM)-based, multi-step pipelines that rewrite and refine the query (e.g., LinCIR, CIReVL) [13, 14]. In parallel, graph-based retrieval explicitly models entities and relations to support structured matching and improve interpretability [8, 15, 16]. However, for personal photo collections, we still lack a single representation that supports contextual querying while also providing visual grounding, traceability, and user revision.

We argue that personal photo management needs representations that support contextual search and fact-grounded narrative construction, while remaining auditable and editable in Human-AI collaboration [17]. Knowledge graphs (KGs) are a natural foundation because they model entities and relations explicitly and support traceable querying [18]. However, existing multimedia KGs often lack pixel-level grounding, temporal awareness for event modelling, and lifecycle mechanisms for correcting AI-generated predictions [18]. In this work, we present PHOTOGRAPH, a claim-centric, spatially and temporally aware KG framework for personal photo understanding and human-AI photobook co-creation. Instead of treating AI outputs as facts, each prediction is stored as a falsifiable *claim* with provenance and, when available, visual evidence. Claims connect persons, objects, scenes, and actions through explicit relations that can be accepted, rejected, or corrected by users. By combining structured representation with grounding and lifecycle tracking, PHOTOGRAPH enables contextual natural language (NL) retrieval with traceable reasoning and supports fact-based visual storytelling. Unlike prior multimedia KGs that store detections as fixed facts, and unlike embedding-based retrieval that returns opaque rankings, the distinguishing feature of PHOTOGRAPH is the combination of evidence-grounded claims, explicit provenance, and a user-driven lifecycle within a single spatially and temporally aware representation, which makes every model output inspectable, correctable, and reversible. We demonstrate KG construction on a sample photo collection and show how event-centric subgraphs support the creation of photobook chapters.

## 2. PhotoGraph

Figure 1 gives an overview of PHOTOGRAPH. The system has three core parts: (1) a set of information extraction (IE) services, (2) a claim-centric KG backbone, and (3) a KG manager which manages the claim lifecycle for KG construction and for human correction and revision tracking. On top of this, PHOTOGRAPH provides downstream modules for (4) retrieval and (5) visual storytelling. When users upload photos, each service produces *claims* with confidence scores and, when available, visual evidence such as bounding boxes. The claim manager merges these outputs, tracks their lifecycle states, and writes accepted claims to the KG, including user corrections.

From an implementation perspective, PHOTOGRAPH is a modular client-server system. The frontend is a React and TypeScript interface that allows for (a) image upload, (b) inspecting and correcting claims, and (c) photobook co-creation. The backend uses FastAPI and an asynchronous job queue

based on Redis to orchestrate extraction pipelines and store claims in a PostgreSQL database. Inside the backend pipelines, the **VLM service** (Qwen3 VL 32B, deployed via `llama.cpp`) extracts objects, actions, and scenes together with bounding boxes. The **face service** (InsightFace `buffalo_l`) performs face detection and embedding-based clustering to suggest identity links. The **geo service** resolves GPS metadata into places and landmarks using the Google Places API. In addition, the **event service** groups images into EVENT structures using temporal and spatial clustering, then refines boundaries using semantic cues already present in the graph, such as shared participants and locations.

## 2.1. Knowledge Graph

Claims are represented as edges in the KG. Each claim carries a predicate and is annotated with provenance (service or model and timestamp), confidence, lifecycle state, and, when available, evidence such as bounding boxes. Users can move a claim from PROPOSED to ACCEPTED, REJECTED, or AUTO ACCEPTED. This keeps uncertain outputs separate from verified facts, while preserving history and accountability.

Conceptually, the graph follows a labeled property-graph (LPG) model rather than an RDF/triple model: claims are first-class, directed edges that carry their own attributes (provenance, confidence, lifecycle state, and evidence), and several services may independently assert the same relation, so parallel edges between the same pair of nodes are permitted (e.g., the two `detected_landmark` edges in Figure 2). We deliberately avoid RDF, where such per-edge metadata and corroboration would require reification or named graphs. The graph is a conceptual layer: physically it is realised over relational tables (separate node, claim, evidence, and user-action tables) in PostgreSQL rather than a dedicated graph store, which keeps lifecycle, provenance, and audit trails under standard relational integrity while preserving property-graph semantics at the model level.

Confidence values are per-claim scores reported by the producing service; we do not currently fuse them into a single global probability. Instead, agreement across independent services is treated as corroboration: when multiple services assert the same relation, the claim becomes eligible for auto-acceptance. A principled aggregation of confidences or likelihoods across heterogeneous services is left to future work.

Figure 2 shows a typical subgraph. The main entity types (IMAGE, PERSON, OBJECT, PLACE, SCENE, EVENT, TIME) are linked by claim edges. The two `detected_landmark` edges to *Bow Bridge* illustrate how evidence from multiple services (VLM and Geo) can support the same relation. The dashed `next_to` edge shows a spatial relation derived from bounding box proximity. Retrieval and storytelling operate on the accepted subgraph (and optionally inferred links) and return outputs together with the supporting claims, so each result comes with a clear justification trace. NL search is handled by translating user queries into structured KG constraints and database queries, which enables contextual and traceable retrieval.

## 2.2. Information Extraction and Graph Integration

Claims are added to the graph incrementally. New outputs reuse existing entities when possible and add new claims instead of overwriting previous information. For visually grounded observations, detections are matched against existing regions (e.g., using IoU overlap) to reduce redundancy while still allowing competing hypotheses to coexist as separate claims. Because competing observations are kept as separate claims rather than merged, contradictions (e.g., two identity hypotheses for one face, or different labels for the same region) are represented explicitly on the shared node or region. Such conflicts are resolved in one of two ways: through the normal claim lifecycle, where a user accepts one claim and rejects the alternatives, or through conflict detection that routes mutually exclusive claims to the human-in-the-loop for adjudication, with rejected alternatives retained for auditing. Face clusters are linked to PERSON entities through editable identity claims. Geo resolution either reuses or creates PLACE entities and keeps `taken_at` claims traceable to their sources. The event module creates EVENT entities from spatio-temporal segments and refines them using cues already present in the KG, such

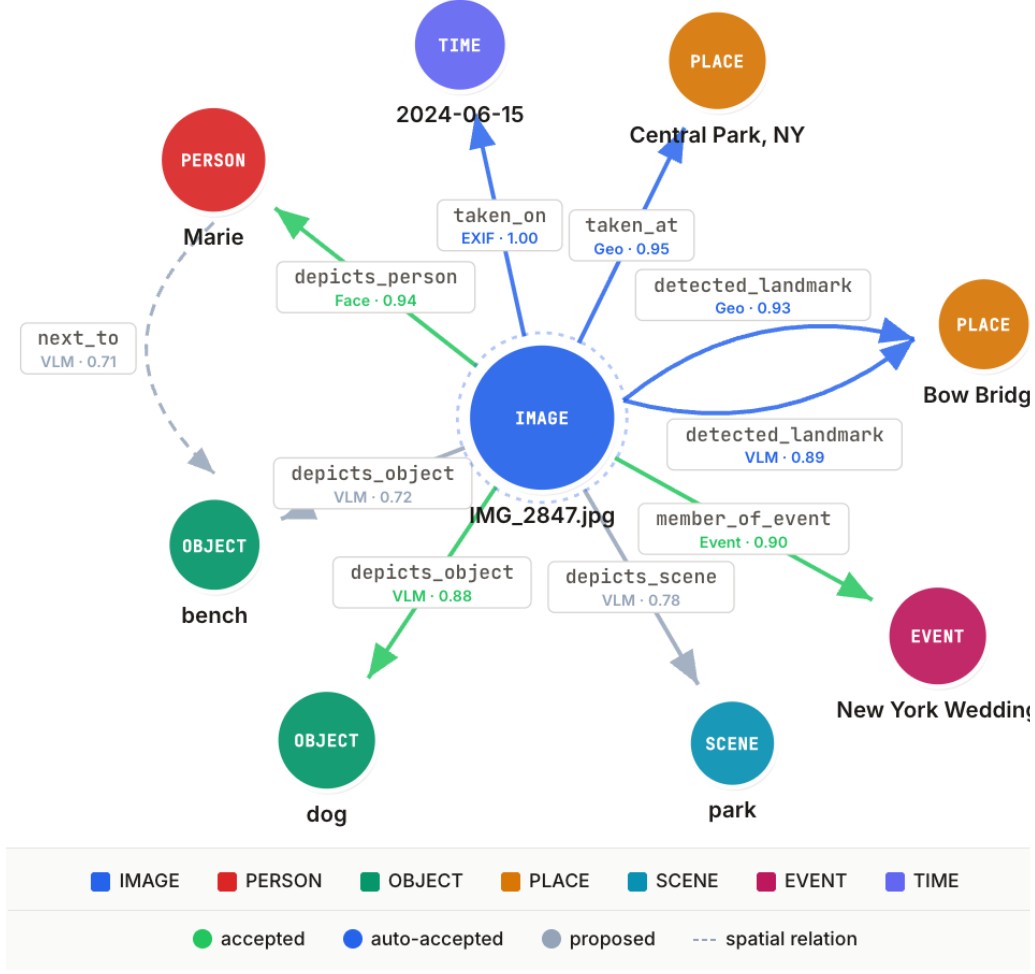

**Figure 2:** Example KG subgraph for a single photograph

as shared people and places. Images are connected to events via membership claims, which supports image retrieval and storytelling while keeping segmentation decisions revisable through the claim lifecycle.

## 2.3. Human-AI Collaboration

A central goal of PHOTOGRAPH is explicit human oversight. Newly generated claims appear in the interface as hypotheses, along with evidence overlays, the producing service or model, and confidence scores. Users can accept, reject, or correct claims, for example, by renaming a person, fixing a location, or removing a false detection. Corrections update the knowledge base without erasing history. Rejected claims remain available for auditing, and corrected claims are linked through lifecycle transitions. This supports iterative curation and reversibility, and helps the system converge to a stable, user-aligned representation of the photo collection.

Initial adoption is a distinct challenge, since importing an existing collection produces many unnamed face clusters at once. To reduce this cost, identity curation operates at the cluster level: naming a face cluster propagates to all of its member claims in a single action, and clusters can be merged or split when the automatic grouping is too coarse or too fine. This lets a user label recurring people in bulk rather than image by image, concentrating manual effort on genuinely ambiguous cases. The same lifecycle then keeps later additions cheap, as high-confidence, multiply-corroborated claims can be auto-accepted while uncertain ones remain proposed and searchable.

## 2.4. Contextual Image Retrieval

To support contextual retrieval, PHOTOGRAPH resolves NL queries directly against accepted claims in the KG using a lightweight three-step pipeline. First, an LLM parser with a regex fallback turns the query into typed constraints over people, objects, places or scenes, actions, tags, and negation, including negations (e.g., not wearing). Second, these constraints are grounded in KG nodes. Named people are resolved via canonical_name to clustered PERSON identities and then linked to per-image instances through the face claim chain, while generic terms such as "woman" use word boundary matching to avoid substring errors. Objects are mapped to detections using labels and categories, places are matched to PLACE nodes and available EXIF fields, and actions are synonym expanded and matched to action claims. Third, the resulting constraints are executed as database patterns over accepted claims, with exclusions handled via NOT EXISTS. Retrieval returns the set of satisfying claims for each result image, providing a justification trace rather than a black box ranking. We currently demonstrate this pipeline on representative multi-constraint queries; a systematic evaluation of parsing and retrieval accuracy over user-generated queries is part of our planned evaluation (Section 3).

## 2.5. Visual Storytelling

PHOTOGRAPH supports photobook chapter creation by organising images into EVENT nodes. Events are formed by segmenting the photo stream over time and then cleaned up using semantic coherence, for example, recurring people and places. Given an event or a chapter selected by the user, the storytelling module picks a small set of representative keyframes and assigns them simple roles such as beginning, peak, and ending based on where they fall within the event timeline. Part of the visual storytelling is generating captions from accepted claims at both the event and image levels. Each caption and narrative element keeps pointers to the supporting claims, so users can see why a sentence was produced and fix the underlying facts when needed.

## 2.6. Demo Interaction

Our demo showcases the full workflow. Users upload a subset of photos, and the system proposes claims with evidence overlays. Users then confirm identities, fix mistakes, and curate the accepted knowledge. Next, they run contextual queries and inspect claim-based explanations for the results. Finally, they generate a photobook chapter from an event, with role-aware captions grounded in accepted facts. Overall, the demo illustrates how a claim-centric KG can serve as a backbone for contextual retrieval and fact-based storytelling, while keeping the process transparent and easy to correct.

## 3. Conclusion & Future Work

PHOTOGRAPH provides a claim-centric, spatially and temporally aware KG for personal photo understanding and photobook co-creation. It supports contextual image retrieval with traceable justifications, and helps users build chapters and storylines from event-centred subgraphs that stay grounded in accepted facts. A limitation is that the overall value of the KG depends on ingestion quality. The representation could be strengthened by richer time and event semantics, better consistency checks across relations, better event extraction [19] and clearer support for unresolved alternatives, for example, multiple possible identities or object types that remain queryable until the user resolves them. From the interaction side, lifecycle-based corrections add effort, and large photo libraries can put pressure on the interface and overwhelm users without careful UI design. Future work will focus on improving ingestion while reducing user effort. Claims can be auto-accepted when confidence is high and corroborated by multiple services, or when the same claim repeats within an event, while uncertain cases remain proposed and searchable. We also plan an evidence-aware judge step that uses structured visual evidence to promote or reject claims, while keeping decisions reversible.

We will also evaluate PHOTOGRAPH against retrieval augmented generation (RAG). RAG can condition generation on retrieved context, but it often relies on unstructured retrieval, provides limited pixel-level

grounding, and offers limited support for persistent correction. A promising direction is KG-grounded RAG, where retrieval returns constraint-satisfying subgraphs and generation is restricted to facts supported by the graph. In its current form, PHOTOGRAPH is a feasibility demonstration rather than a quantitatively evaluated system: we show that the claim-centric KG supports contextual retrieval and storytelling, but we have not yet measured retrieval quality at scale. Our planned evaluation therefore targets three open questions raised during review: (i) the accuracy with which the LLM parser and grounding pipeline map free, user-generated natural-language queries to the intended photos, measured on a held-out query set rather than scripted examples; (ii) retrieval efficiency and quality as the library grows to thousands of images, particularly for queries that combine multiple people, places, and negation; and (iii) a head-to-head comparison with unstructured and KG-grounded RAG baselines. We will evaluate PHOTOGRAPH in user studies and test it at scale across larger, more diverse datasets. Beyond photobooks, the same approach could support audit-critical visual archives, such as medical collections and security or forensic data.

## Acknowledgments

This work was funded by the Federal Ministry of Research, Technology and Space (BMFTR) under grant number 16IW23002 (No-IDLE) and grant number 16IW24006 (NoIDLEChatGPT), the Lower Saxony Ministry of Science and Culture (MWK) in the zukunft.niedersachsen program, and the Endowed Chair of AAI at the University of Oldenburg.

## Declaration on Generative AI

During the preparation of this work, the authors used Claude (Claude Opus 4.8) in order to: paraphrase and reword, and improve the writing style. After using this tool, the authors reviewed and edited the content as needed and take full responsibility for the publication's content. Generative models additionally appear as components of the system described in this paper (the Qwen3-VL service and the LLM query parser, Section 2); these are part of the proposed method rather than the manuscript-writing process.

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
