# OpenReview forum: "PhotoGraph: Claim-Centric Knowledge Graphs for Personal Photo Management"
_ijcai.org/IJCAI-ECAI/2026/Workshop/GENAIK-NORA — IJCAI-ECAI 2026 Joint Workshop on GENAIK and NORA_

### Official Review · Reviewer_KS2D · 2026-05-17
**Claim-Centric Knowledge Graph Framework for Personal Photo Understanding and Photobook Co-Creation**

**Rating:** 7
**Confidence:** 4

**Review:**

This paper introduces a new claim-centric, spatially and temporally aware knowledge graph framework for personal
photo understanding and photobook co-creation that is based on Human-AI collaboration. Authors argue that knowledge graph structure as a unified representation of different aspects of photos better suits the task researched than several separate and unconnected representations of entities and their relations used in some previous works. To produce solid evidence they propose to reply on probabilistic claims rather than hard facts. The proposed framework is composed of several services responsible for object detection, face detection and geolocation identification. Technical implementation is sufficiently well described to facilitate its understanding by more technical readers. Initial experiments with few photos as well as attached video demonstrate feasibility of this framework.

What is omitted from the paper, in my opinion, is

1) How different is this framework from others for the chosen task? A few words were given in the paper but still this may be not enough to judge of its advantages and disadvantages.
2) Claims are likely probabilistic in their nature. How do you combine probabilities or likelihoods?

Overall, I warmly recommend this paper for acceptance due to its practical importance and obtained results as well as well-structured description.

---

### Official Review · Reviewer_Xban · 2026-05-21

**Rating:** 5
**Confidence:** 3

**Review:**

Paper Summary:

This paper proposes a claim-centric, spatially and temporally aware knowledge graph framework for personal photo management and photobook co-creation. Instead of treating AI-generated detections as fixed facts, the system stores them as evidence-grounded claims with confidence scores, provenance, lifecycle states, and optional visual grounding such as bounding boxes. These claims connect people, objects, scenes, places, events, and times, enabling users to inspect, accept, reject, or correct model outputs. The resulting knowledge graph supports contextual natural-language image retrieval with traceable justifications, as well as event-based visual storytelling for generating photobook chapters.

Pros:
1. This paper has a clear motivation in practice.
2. This paper is well-organized, and the writing is easy to follow.

Cons:
1. The proposed framework allows competing hypotheses to coexist, but it is unclear how contradictions are represented and resolved.
2. The conclusion states that PhotoGraph will be evaluated against RAG, but given the paper’s central argument that KG-grounded representations are preferable to unstructured retrieval, such a comparison should ideally be included in the current work.
3. The novelty of this paper is a bit limited. It reads mostly like a combination of existing ideas.

---

### Official Review · Reviewer_HXpR · 2026-06-05
**Helpful approach to photo management that would benefit from evaluation data**

**Rating:** 5
**Confidence:** 4

**Review:**

This paper presents an approach to personal photo collection management that uses knowledge graphs to support accurate retrieval of complex user requests. By structuring information in and about the images, users are able to query for more involved scenarios such as `Marie wearing a black jacket at the beach’.

The project connects images to knowledge graphs including temporal, event, and entity information. The design of the graph, the mapping of images to the graph, and the user interface are the primary contributions. The authors provide a video that shows the user interface and the general functionality of the PhotoGraph tool. It is a clean interface where the user can see what needs to be done at each step and it shows the results of several queries, all of which look very promising.

There are two areas that are only addressed briefly in the paper that seem fundamental to the main claims. One is the evaluation of the performance of the tool and the other is barriers to initial adoption of the tool.

It is not clear how robust the system is to user generated queries. Since being able to query the system with natural language is one of the major contributions, it would be good to see evidence that it correctly maps natural language queries to the right photos. The video demo shows examples that are stated in the text or very similar to those, so it serves as a proof of concept, but does not support the claim that the tool can generally handle queries with multiple constraints. It would be good, for instance, to have an appendix of user-generated queries along with an evaluation of how accurately the tool responded, or some other metric showing that the demo examples are not isolated working instances.

It would also be good, on the evaluation point, to understand how the system scales to a large number of photos. Does the combination of multiple locations and entities with negation allow efficient retrieval?

Regarding initial adoption of the tool, the authors might address ways to make it easier to set it up on a personal collection. Changing to a new tool, people are going to often have thousands of photos all at once, even in personal collections, which can make adoption of the tool harder. The authors mention the idea of automatically labelling things to help with the user experience. That seems like a good route for existing users. As the user adds photos, it can, as the authors suggest, automatically add high enough confidence instances to the knowledge graph.  But it isn't clear how this approach would work for the initial set up. For example, people detected in images in their private collection will be separated into different sets, but the user will have to provide the names for the sets for them to be meaningful in queries. The same holds for specific locations, such as “Maria’s house”. Having a plan around onboarding people is important to address.

The work that has been done represents a good approach to photo management, but stands as a proof of concept regarding the use case, due to the lack of systematic evaluation of its question answering capabilities.

---

### Official Review · Reviewer_Y9G1 · 2026-06-09
**An interesting application case study for many important problems in information extraction and multi modal KGs**

**Rating:** 7
**Confidence:** 5

**Review:**

The demo paper is interesting to read. I found the design choice of modelling claims as arcs between entities, annotated with provenance, particularly interesting. From the example, it appears the KG backend is a property graph rather than an RDF graph; the authors should clarify the KG meta-model of reference, as there are important differences (e.g., duplicated arcs are not allowed in RDF), and motivate their decision. I think the demo is an interesting application case study for many important problems in information extraction and multi-modal KGs. However, I struggle to see people curating image collections in that way for personal use. I recommend that the authors to look at more compelling application domains, for example, cultural heritage museum collections, like photography archives, for which such an workflow is of crucial importance.

---

### Decision · Program_Chairs · 2026-06-10

Accept